# Exploring the Coordinated Development of Smart-City Clusters in China: A Case Study of Jiangsu Province

Guoqing Shi [1,2,3,*], Bing Liang [1,2,3,*], Taotao Ye [4], Kexin Zhou [1,2,3] and Zhonggen Sun [1,2,3]

1 National Research Center for Resettlement, Hohai University, Nanjing 211100, China; zkx9808zkx@163.com (K.Z.); sunzhonggen@sina.com (Z.S.)
2 Institute of Social Development, Hohai University, Nanjing 211100, China
3 Asian Research Center, Hohai University, Nanjing 211100, China
4 Department of Economics and Manageent, Jiangsu Provincial Committee Communist Party College, Nanjing 210009, China; y907994533@163.com
* Correspondence: gshi@hhu.edu.cn (G.S.); bliang1@126.com (B.L.)

**Abstract:** As urbanization has accelerated, China has started to build smart cities, which have formed smart-city clusters. It is critical to coordinate development within smart-city clusters to enhance the efficiency of city-cluster construction. From the perspective of demographic economics, this study innovatively constructed an evaluation system for the coordinated development of smart-city clusters and utilized the coupled coordination degree model to conduct an in-depth study of smart-city clusters in Jiangsu Province. The results show that there are clear differences in the development between the three regions of Jiangsu Province: Southern Jiangsu, Central Jiangsu, and Northern Jiangsu. The development within Jiangsu Province is imbalanced, where the overall development trend is high in the southern region and low in the northern region. The main driving factors include geography, the Matthew effect, game thinking, and industrial structure. Accordingly, the results suggest the following recommendations for the coordinated development of smart-city clusters: strengthening cross-regional cooperation, promoting data sharing and interoperability, deepening synergistic industrial development, and expanding innovation capacity.

**Keywords:** smart-city cluster; coordinated development; evaluation index system; coupled coordination degree model; Jiangsu Province



## 1. Introduction

With the rapid acceleration of urbanization and the rapid development of information technology in China, the construction of smart cities has become an important strategic choice to promote high-quality urban development. Smart cities, characterized by informatization, intelligence, and green low-carbon features, aim to enhance urban operational efficiency, improve residents' quality of life, and promote sustainable economic and social development [1–7]. However, the construction of smart cities in isolation is no longer sufficient to meet the needs of urban development, highlighting the importance of interconnection and coordinated development among cities.

In recent years, there has been a surge of smart-city construction across China, with many cities accelerating the pace of smartization and witnessing the emergence of numerous smart-city-demonstration zones and innovative practices. Concurrently, coordinated development among city clusters has become an important path for China's urbanization [8]. The formation of city clusters aims to optimize resource allocation, promote industrial coordination, enhance overall economic benefits and competitiveness, and achieve complementarity and linkage among cities [9]. However, challenges such as information silos, inconsistent technical standards, and inadequate policy frameworks often hinder the coordinated development of smart-city clusters [10].

Therefore, delving into the paths and mechanisms for the coordinated development of smart-city clusters in China holds significant theoretical and practical significance. By planning the smart-city clusters comprehensively and constructing collaborative innovation mechanisms, cities can better leverage their comparative advantages, realize resource sharing and complementarity, and promote the sustainable development of city clusters [11]. Additionally, actively exploring the application of smart technologies in urban governance, public services, and other areas can enhance the overall operational efficiency of city clusters and improve the quality of life for residents, thus creating more harmonious and livable urban-cluster spaces [12].

Given the context of China's rapid urbanization and the burgeoning development of smart cities, this study aims to investigate the following question: how can the coordinated development of smart-city clusters in China be effectively explored and promoted, considering the challenges and opportunities in terms of information integration, technical interoperability, policy alignment, and sustainable development? We take Jiangsu Province as an example to study the balanced development of smart-city clusters in the region. This study aims to scientifically assess regional development, which provides a basis for domestic and international government departments to formulate regional development policies in a scientific and reasonable manner and theoretical references for academic research on balanced regional development.

## 2. A Review of the Evaluation of the Balanced Development of Smart-City Agglomeration

China has been promoting coordinated regional development in recent years, focusing on equity and efficiency to achieve the goal of balanced and high-quality development across all regions [13]. In 2010, Jiangsu Province proposed a strategy for coordinated regional development in 2010 to narrow the regional development gap. An important objective of balanced regional development is the balance between regional coordination [14] and regional development. While there is no common definition of regional coordinated development in academia, its connotation has evolved with social practice. The focus of regional coordinated development in China lies in coordinating economic development. A report from the Twentieth National Congress of the CPC noted that "the state ensures that the national economy can realize coordinated development in proportion to its size through the auxiliary role of market regulation and the comprehensive balance of economic planning." With the deepening of the reform and opening up, the economic and social development of China has become unbalanced, and the level of development between different regions has gradually widened [15]. These disparities have led to many social problems, pushing the balanced development of regions to be one of the important goals on the government's agenda. The quantitative evaluation of the level of regional coordinated development is an important part of applying theoretical research to social practice [16]. To create an index system [17], Chinese scholars mainly adopt three methods: theoretical analysis, frequency statistics, and principal component analysis [18]. Such an index system usually takes on the form of either a hierarchical index system or a comprehensive index system [19,20]. From a spatial perspective, researchers have analyzed at the country, geographic region, urban agglomeration, and administrative region levels, considering a variety of factors [21–24]. To evaluate the coordinated development of regions in China, Xu Yingzhi and Wu Haiming used the "four-point method," by constructing four evaluation subsystems of society, economy, ecology, and science and technology, covering ten levels of criteria and a total of 27 evaluation indices [25]. Zhang Chao constructed a regional coordinated development evaluation index system that encompassed five components, namely, public services, infrastructure, economic development, the ecological environment, and quality of life, and measured the regional coordinated development level of 30 provinces in China from 1996 to 2017 [26]. Song Shengnan constructed eight first-level indices and 27 s level indices of industrial structure, spatial connection, market development, infrastructure, public service, environmental protection, urban-rural integration, and development mecha-

nisms to evaluate the regional coordinated development level of the Hefei Metropolitan Area [27]. These studies constructed different evaluation index systems depending on their study area and purpose, but the selection of the all the indices commonly reflects the concepts of hierarchy, diversity, and complexity. Population mobility highlights the comparative advantages of regional development [28], while economic development provides the material conditions for coordinated regional development [29]. However, studies on coordinated regional development have not considered the influence of demographic and economic factors in designing an index system [30].

The idea of a demographic economy originated with the emergence of mercantilists in the 16th century. Classical economists, including François Quesnay and William Peddie, considered demographic factors in the field of economics. These authors first discussed the relationship between wealth and population [31]. In subsequent studies, scholars such as David Ricardo, Thomas Robert Malthus, and Adam Smith proposed the theory of demographic economics based on the law of diminishing land harvest and the theory of the value of labor [32]. After the 20th century, the study of Western demographic economics formed a neoclassical economic theoretical framework based on macroeconomics and microeconomics, which included the theory of neoclassical economics [33], the push–pull model, the quadratic exponential model [34], and Lee's migration theory [35]. Compared with those in Western countries, research on population economics in China began much later. After the reform and opening up, Chinese researchers and scholars systematically studied the population economics theory by drawing on relevant theories from Western population economics [36]. Many scholars have studied regional coordinated development from the perspectives of the economy, population, resources, and the environment [37]. However, most studies focus on regional sustainable development from the perspective of the environmental sciences [38], and few studies address regional coordinated development by combining demographic and economic factors from the perspective of the social sciences [39].

In regional balanced development, the local population, economy, and social development[1] are important factors and measurement indices. Therefore, this study constructs an evaluation index system from these three perspectives, which includes a development index and a degree of coordination index, to judge whether regional development is balanced.

## 3. Methods and Data Sources

### 3.1. Construction of the Evaluation Index System

Building an index system is a core component of systematic evaluation and thus affects the reliability of the evaluation results [40]. The regional population–economic–social development evaluation index system constructed in the study follows the following four principles [41]: (1) Representativeness: the selected indices must reflect the situation and characteristics of regional population-economic-social development to the maximum extent possible; (2) Operability: the indices should be selected in a simple and clear way in which the data are easy to collect and calculate; (3) Scientific Rigor: the indices must adopt scientific calculation methods; (4) Systematicity: there should be a certain degree of logic between indices that reflects the main characteristics of the demographic–economic–social subsystems from different aspects.

#### 3.1.1. Construction of the Evaluation Index System for the Population-Development Subsystem

The population-development subsystem is complex and contains many elements [42]. Under the primary index of the population development subsystem, this study constructed three secondary indices: population size, population quality, and population structure. The demographic factor itself has both natural and social attributes, corresponding to both the quantity and quality of the population. Population scale is then divided into four three-level indices: the number of births, the number of deaths, the household population at the end of the year, and the resident population at the end of the year. The quality of

the population refers to the education level of the population, which is captured through three tertiary indices: the illiteracy and semi-illiteracy rate, the percentage of people with a Bachelor's degree or higher, and the percentage of people with a high-school education or lower. The population structure usually includes gender structure, age structure, and urban–rural structure, and encompasses six three-level indices: sex ratio at birth, ratio of 0–14-year-olds, ratio of 15–64-year-olds, ratio of 65-year-olds and above, population density, and ratio of the urban population. The results of the construction of the evaluation index system for the population-development subsystem are shown in Table 1.

**Table 1.** System of indices for evaluating the population-development subsystem.

| Target Level | Standardized Layer | Serial Number | Index Layer | Index Properties |
|---|---|---|---|---|
| Population-development subsystem indices | Size of population | A1 | Number of births (persons) | + |
| | | A2 | Number of deaths (persons) | − |
| | | A3 | Household population at the end of the year (10,000) | + |
| | | A4 | Year-end resident population (10,000) | + |
| | Quality of population | A5 | Illiteracy and semi-illiteracy (%) | − |
| | | A6 | Percentage of people with Bachelor's degree or above (%) | + |
| | | A7 | Percentage of people with high-school education or less (%) | − |
| | Population structure | A8 | Sex ratio at birth | − |
| | | A9 | Percentage of 0–14-year-olds (%) | + |
| | | A10 | Percentage of 15–64-year-olds (%) | + |
| | | A11 | Percentage of persons aged 65 and over (%) | + |
| | | A12 | Population density (persons/km$^2$) | + |
| | | A13 | Percentage of urban population (%) | + |

### 3.1.2. Construction of an Evaluation Index System for the Economic-Development Subsystem

Regional economic development includes both the quantitative and qualitative aspects of the economic development of regions [43]. Under the first-level index of the economic-development subsystem, the study constructs three second-level indices: economic scale, economic quality, and economic structure. Three third-level indices are divided under the second-level index of economic scale: year-end gross domestic product, general public budget revenue, and total industrial output value. These indices were selected to reflect the dimension of social activities and public life in a region from an economic perspective. Four tertiary indices were selected under the second-level index of economic quality: per capita GDP, GDP growth rate, per capita local financial revenue, and per capita gross industrial output value. The evaluation index system for the economic-development subsystem are shown in Table 2.

**Table 2.** System of indices for evaluating the economic-development subsystem.

| Target Level | Standardized Layer | Serial Number | Index Layer | Index Properties |
|---|---|---|---|---|
| Economic-development subsystem indices | Size of the economy | B1 | Year-end gross domestic product GDP (billions of yuan) | + |
| | | B2 | General public budget revenue (billions of yuan) | + |
| | | B3 | Gross industrial output (billions of yuan) | + |

| Target Level | Standardized Layer | Serial Number | Index Layer | Index Properties |
|---|---|---|---|---|
| | | B4 | GDP per capita (yuan) | + |
| | | B5 | GDP growth rate (%) | + |
| | Quality of the economy | B6 | Per capita local fiscal revenue (million yuan) | + |
| | | B7 | Per capita gross industry output (yuan) | + |
| | | B8 | Share of primary production value (%) | + |
| | Economic structure | B9 | Share of secondary production value (%) | + |
| | | B10 | Share of output value of the third sector (%) | + |

### 3.1.3. Construction of an Evaluation Index System for Social-Development Subsystems

Social development generally includes many elements, such as politics, culture, ecology, and economy [44]. In this study, the elements of social development exclude the other two factors of the population and economy [45]. Under the first-level index of the social-development subsystem, there are four second-level indices: infrastructure, culture and education, the medical system, and the ecological environment. Under the second-level index infrastructure, there are four third-level indices: road area per capita, public transportation vehicles per 10,000 people, Liquefied petroleum gas supply per 10,000 people, and water supply per capita. Under the second-level index culture and education, there are four tertiary indices: per capita financial expenditure on education, total number of students at each stage of schooling, total number of teachers at each stage of schooling, and number of books in public libraries per capita. There are five tertiary indices under the secondary index of healthcare: total number of hospitals and health centers, number of hospital beds per 10,000 people, percentage of people enrolled in basic medical insurance, percentage of people enrolled in work-related injury insurance, and percentage of people enrolled in worker's unemployment insurance. The ecological environment is captured by three tertiary indices: sewage-treatment rate, amount of domestic garbage removal, and per capita green space in parks. The results of constructing the evaluation index system for the social-development subsystem are shown in Table 3.

**Table 3.** System of indices for evaluating the social-development subsystem.

| Target Level | Standardized Layer | Serial Number | Index Layer | Index Properties |
|---|---|---|---|---|
| | | C1 | Road area per capita (square meters) | + |
| | | C2 | Public transportation vehicles for 10,000 people (standard units) | + |
| | Infrastructure | C3 | Supply of liquefied petroleum gas per 10,000 people (tons) | + |
| | | C4 | Water supply per capita (tons) | + |
| | | C5 | Per capita financial expenditure on education (yuan) | + |
| | | C6 | Total number of students enrolled in school at all levels (10,000) | + |
| Social-development subsystem indices | Cultural Education | C7 | Total number of teachers at all levels (10,000) | + |
| | | C8 | Public library holdings per capita (volumes) | + |
| | | C9 | Total number of hospitals, health centers | + |
| | Medical System | C10 | Number of hospital beds per 10,000 persons (sheets) | + |
| | | C11 | Percentage of persons covered by basic health insurance (%) | + |

| Target Level | Standardized Layer | Serial Number | Index Layer | Index Properties |
|---|---|---|---|---|
| | Medical System | C12 | Percentage of employees insured against work-related injuries (%) | + |
| | | C13 | Unemployment insurance participation (%) | + |
| | | C14 | Sewage-treatment rate (%) | + |
| | Ecological Environment | C15 | Volume of domestic waste removed (tons) | + |
| | | C16 | Green space per capita in parks (square meters) | + |

*3.2. Construction of the Coupled and Coordinated Development-Evaluation Model*

The Coupled Coordinated Development-Evaluation Model (CCDEM) is a comprehensive analytical framework designed to assess and evaluate the interconnectedness and harmony among various components within a system or between multiple systems. This model is particularly valuable in the context of urban and regional development, where the coupling and coordination of different factors play a crucial role in achieving sustainable and balanced growth. At its core, the CCDEM considers multiple dimensions of development, including economic, social, environmental, and technological factors. By quantitatively measuring the coupling strength and coordination level among these dimensions, the model provides insights into the overall efficiency and effectiveness of development strategies. The Coupled Coordinated Development-Evaluation Model (CCDEM) is an instrumental methodology for investigating the coordinated advancement of smart-city clusters in China. This model is highly relevant to the research theme as it offers a systematic framework aligning seamlessly with the overarching aim of fostering synchronized and sustainable urban growth. In summary, the Coupled Coordinated Development Evaluation Model (CCDEM) presents a robust analytical framework for scrutinizing and steering the synchronized progress of smart-city clusters in China. Its multidimensional approach, capacity for comparative analyses, and dynamic nature harmonize adeptly with the complexities and exigencies inherent in cultivating sustainable and equitable urban development within the realm of smart-city clusters.

3.2.1. Determination of Weights Using the Entropy Weight Method (EWM)

When choosing methods to measure the level of coordinated development of a region, we chose to use the entropy weight method [46], which is relatively accurate and objective. The steps involved in this method are as follows.

Step 1: Construct the initial judgment matrix. The evaluation target of region $n$ for year $r$ is s, and there are a total of $m$ evaluation indices; then, there is a judgment matrix.

$$A = \left\{ X_{qij} \right\} (q = 1, 2 \cdots r; i = 1, 2 \cdots n; j = 1, 2 \cdots m) \tag{1}$$

Step 2: Conduct a dimensionless quantization of indices. Since the selected indices contain positive and negative indices, each index has a different magnitude and unit. To facilitate uniform calculation, the indices are made dimensionless using the method of polarity transformation. A new matrix is obtained:

$$A^* = \left\{ x_{qij}^* \right\} \tag{2}$$

Positive indices:

$$x_{qij}^* = \left( x_{qij} - \min\left\{ x_{qij} \right\} \right) / \left( \max\left\{ x_{qij} \right\} - \min\left\{ x_{qij} \right\} \right) \tag{3}$$

Negative indices:

$$x^*_{qij} = \left(\max\{x_{qij}\} - x_{qij}\right) / \left(\max\{x_{qij}\} - \min\{x_{qij}\}\right) \tag{4}$$

Step 3: Calculate the entropy weight of the $j$-th index in the $q$-th year $e_j (0 \le e_j < 1)$. $P_{qij}$ is the weight of the $j$-th index, $k = 1/\ln(m)$, $k > 0$, and $m$ is the number of indices

$$p_{qij} = x^*_{qij} / \sum_q \sum_i x^*_{qij} \tag{5}$$

$$e_j = k \sum_q \sum_i p_{qij} \ln\left(p_{qij}\right) \tag{6}$$

Step 4: Calculate the coefficient of variation ($d_j$) for index (column) $j$.

$$d_j = 1 - e_j \tag{7}$$

Step 5: Calculate the weight of the $j$-th index (column).

$$w_j = d_j / \sum d_j \tag{8}$$

Step 6: Calculate the final evaluation target weights for region $i$

$$S_{gk} = \sum_j w_j x_{qij} \tag{9}$$

3.2.2. Modeling the Degree of Coordination of Coupled Demographic–Economic–Social Systems

The coupling-coordination degree model is used to analyze the level of coordinated development [47]. The coupling degree refers to the degree of mutual influence between two or more systems and can reflect the degree of connection between systems. The degree of benign coupling development between coordination degree subsystems can reflect the degree of coordination between systems. The study constructs the coupling-coordination degree model of the population–economy–society (PES) integrated system to judge the coordination level status of social development in district cities. The standardized population, economic, and social system indices and the multiplication of the weights of each index can be accumulated to obtain the score of each subsystem, which is denoted as $\alpha$, $\beta$, and $\lambda$, respectively.

PES (population–economy–society)-system-development coupling index:

$$C1 = \sqrt[3]{\frac{\alpha\beta\lambda}{\left(\frac{\alpha+\beta+\lambda}{3}\right)^3}} \tag{10}$$

To measure the degree of coordination between subsystems, a development index $T$ and a coordination index $D$ are introduced.

Calculation of the PES development index ($T_{ijk}$) and coordination index (named $D_{ijk}$):
Development indices:

$$T_{ijk} = \theta \times A + \zeta \times B + \gamma \times C \left(\theta = \zeta = \gamma = \frac{1}{3}; A, B, C \in (\alpha, \beta, \lambda)\right) \tag{11}$$

Coordination index:

$$D_{ijk} = \left(C_{ijk} \times T_{ijk}\right)^{\frac{1}{2}} \tag{12}$$

According to previous relevant studies [48–50], the classification criteria of the development indices adopted in this study are shown in Table 4, and the coupling-coordination sdegree-level classification criteria are shown in Table 5.

**Table 4.** Criteria for classifying development indices.

| Development Index | [0–0.3) | [0.3–0.4) | [0.4–0.5) | [0.5–0.6) | [0.6–0.7) | [0.7–0.8) | [0.8–1] |
|---|---|---|---|---|---|---|---|
| Level of development | Extremely low | Medium low | Lower | Medium | Higher | Medium high | Extremely high |
| | | Low | | Medium | | High | |

**Table 5.** Classification of the coupling-coordination degree.

| Coordination Index | Coordination Phase | Degree of Coordinated Development |
|---|---|---|
| [0–0.1) | | Extremely disordered |
| [0.1–0.2) | Disordered type | Severely disordered |
| [0.2–03) | | Mildly disordered |
| [0.3–0.4) | | Endangered coordination |
| [0.4–0.5) | Transition type | Fragile coordination |
| [0.5–0.6) | | Barely coordinated |
| [0.6–0.7) | | Basic coordination |
| [0.7–0.8) | | Intermediate coordination |
| [0.8–0.9) | Coordinated development | Well-coordinated |
| [0.9–1.0] | | High-quality coordination |

*3.3. Study Area and Data Sources*

Jiangsu Province is located in the center of the eastern coast of mainland China, with an area of approximately 107,200 square kilometers, which accounts for 1.12% of the country's total area. The province also has the smallest per capita land area in China. The terrain of Jiangsu Province is dominated by plains, with the proportion of plains being the highest in the country. Jiangsu Province governs a total of 13 district cities. According to the geographical location of its district-level cities, Jiangsu Province can be divided into three regions: Southern Jiangsu, Central Jiangsu, and Northern Jiangsu. South of the Yangtze River is the Southern Jiangsu region, north of the Huai River is the Northern Jiangsu region, and between the Yangtze River and the Huai River is the Central Jiangsu region. The Southern Jiangsu region includes the five cities of Nanjing, Suzhou, Wuxi, Changzhou, and Zhenjiang; the Jiangsu region includes the three cities of Yangzhou, Taizhou, and Nantong; and the Northern Jiangsu region includes the five cities of Xuzhou, Lianyungang, Suqian, Huai'an, and Yancheng. Please refer to Figure 1 (Location map of Jiangsu Province divided into southern, central, and northern Jiangsu Province) for details. These three regions are characterized not only by geographical and cultural differences but also by significant differences in their levels of economic development. Therefore, Jiangsu Province is selected as the research object for regional coordinated development because it is diverse and representative. This research provides some reference and guidance for research on regional coordinated development strategies in Jiangsu Province, in China and even worldwide.

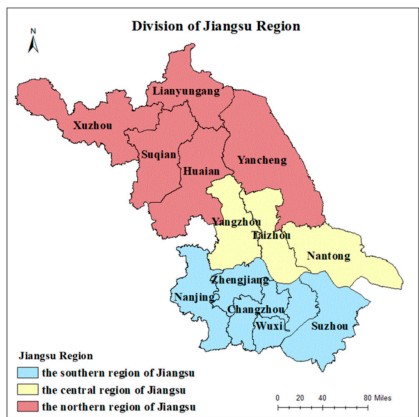

**Figure 1.** Location map of Jiangsu Province divided into southern, central, and northern Jiangsu Province.

The research data come from the annual statistical yearbooks published on the official website of the Jiangsu Provincial Bureau of Statistics. To study the balanced regional development of Jiangsu Province during the 12th and 13th Five-Year Plan periods, the statistical data from 2010, 2015, and 2020 were selected for analysis.

## 4. Results

### 4.1. Index Weighting Values

The final results of the weight values of the indices for the demographic, economic, and social systems for 2010, 2015, and 2020 were calculated using the entropy weighting method and are shown in Table 6.

**Table 6.** Weights of demographic–economic–social-system indices (2010, 2015, 2020).

| Level 1 Indices | Secondary Indices | Tertiary Indices | $W_{2010}$ | $W_{2015}$ | $W_{2020}$ |
|---|---|---|---|---|---|
| Population-development subsystem indices | Size of population | Number of births (persons) | 0.09 | 0.08 | 0.07 |
| | | Number of deaths (persons) | 0.05 | 0.06 | 0.05 |
| | | Household population at the end of the year (10,000) | 0.06 | 0.06 | 0.06 |
| | | Year-end resident population (10,000) | 0.08 | 0.07 | 0.09 |
| | Quality of population | Illiteracy and semi-illiteracy (%) | 0.07 | 0.10 | 0.06 |
| | | Undergraduate education and above (%) | 0.07 | 0.10 | 0.06 |
| | | High-school education and below (%) | 0.08 | 0.06 | 0.07 |
| | | sex ratio at birth | 0.08 | 0.08 | 0.05 |
| | Population Structure | Percentage of 0–14-year-olds (%) | 0.08 | 0.07 | 0.07 |
| | | Percentage of 15–64-year-olds (%) | 0.12 | 0.10 | 0.12 |
| | | Percentage of persons aged 65 and over (%) | 0.04 | 0.05 | 0.07 |
| | | Population density (persons/km2) | 0.07 | 0.09 | 0.11 |
| | | Percentage of urban population (%) | 0.10 | 0.09 | 0.11 |
| Economic-development subsystem indices | Size of economy | Year-end GDP (billions of yuan) | 0.11 | 0.13 | 0.12 |
| | | Public budget revenue (billions of yuan) | 0.12 | 0.14 | 0.15 |
| | | Gross industrial output (billions of yuan) | 0.11 | 0.12 | 0.12 |
| | Quality of the economy | GDP per capita (yuan) | 0.09 | 0.10 | 0.08 |
| | | GDP growth rate (%) | 0.05 | 0.05 | 0.07 |
| | | Per capita local fiscal revenue (ten thousand yuan) | 0.09 | 0.09 | 0.11 |
| | | Gross industrial output per capita (million yuan) | 0.10 | 0.12 | 0.10 |
| | Economic Structure | Share of primary production value (%) | 0.11 | 0.12 | 0.11 |
| | | Share of secondary production value (%) | 0.10 | 0.05 | 0.04 |
| | | Share of output value of the third sector (%) | 0.12 | 0.06 | 0.10 |
| Social-development subsystem indices | Infrastructure | Road area per capita (square meters) | 0.05 | 0.05 | 0.08 |
| | | Public transportation vehicles for 10,000 people (standard units) | 0.06 | 0.04 | 0.07 |
| | | Oil and gas supply for 10,000 people (tons) | 0.05 | 0.06 | 0.04 |
| | | Water supply per capita (tons) | 0.07 | 0.08 | 0.07 |
| | Cultural Education | Per capita financial expenditure on education (yuan) | 0.06 | 0.06 | 0.06 |
| | | Total number of students in school (10,000) | 0.04 | 0.06 | 0.07 |
| | | Total number of teachers at all stages (10,000) | 0.05 | 0.05 | 0.06 |
| | | Public library holdings per capita (volumes) | 0.06 | 0.07 | 0.06 |
| | Medical Protection | Total number of hospitals, health centers (number) | 0.04 | 0.04 | 0.04 |
| | | Number of beds per 10,000 people (beds) | 0.07 | 0.06 | 0.03 |
| | | Number of people enrolled in basic health insurance (%) | 0.06 | 0.07 | 0.06 |
| | | Number of persons insured against work-related injuries (%) | 0.08 | 0.09 | 0.09 |
| | | Number of participants in unemployment insurance (%) | 0.07 | 0.07 | 0.08 |
| | Ecological Environment | Sewage-treatment rate (%) | 0.08 | 0.03 | 0.07 |
| | | Volume of domestic waste removed (tons) | 0.11 | 0.12 | 0.09 |
| | | Per capita green space in parks (square meters) | 0.05 | 0.03 | 0.04 |

### 4.2. Development Indices

According to the formula of the development index, the PES-development indices of the 13 cities in Jiangsu Province in 2010, 2015, and 2020 were obtained. According to

the classification criteria of the development indices in Table 4, the development levels of the 13 cities in Jiangsu Province in the corresponding years were determined, as shown in Table 7. Combined with the spatial–temporal evolution of the development indices of the PES system in Jiangsu Province, see Figure 2 (Distribution of the PES-system-development indices of each prefecture-level city in Jiangsu Province from 2010 to 2020.) for details, the following conclusions can be drawn.

**Table 7.** List of PES comprehensive population–economic–social-development indices for prefectural-level cities in Jiangsu Province.

| Regions | | 2010 | | | 2015 | | | 2020 | | |
|---|---|---|---|---|---|---|---|---|---|---|
| | | Index | Rankings | Leve | Index | Rankings | Leve | Index | Rankings | Leve |
| Southern Jiangsu | Suzhou | 0.71 | 1 | Medium–High | 0.77 | 1 | Medium–High | 0.76 | 1 | Medium–High |
| | Nanjing | 0.65 | 2 | Higher | 0.70 | 2 | Medium–High | 0.72 | 2 | Medium–High |
| | Wuxi | 0.63 | 3 | Higher | 0.60 | 3 | Higher | 0.59 | 3 | Higher |
| | Changzhou | 0.45 | 4 | Lower | 0.45 | 4 | Lower | 0.46 | 4 | Lower |
| | Zhenjiang | 0.36 | 6 | Medium–Low | 0.39 | 6 | Medium–Low | 0.37 | 6 | Medium–Low |
| Central Jiangsu | Nantong | 0.40 | 5 | Lower | 0.42 | 5 | Lower | 0.40 | 5 | Lower |
| | Yangzhou | 0.35 | 7 | Medium–Low | 0.34 | 8 | Medium–Low | 0.33 | 8 | Medium–Low |
| | Taizhou | 0.29 | 9 | Extremely Low | 0.28 | 10 | Extremely Low | 0.30 | 9 | Medium–Low |
| | Xuzhou | 0.33 | 8 | Medium–Low | 0.36 | 7 | Medium–Low | 0.32 | 7 | Medium–Low |
| Northern Jiangsu | Yancheng | 0.26 | 10 | Extremely Low | 0.29 | 9 | Extremely Low | 0.27 | 10 | Extremely Low |
| | Huai'an | 0.24 | 11 | Extremely Low | 0.28 | 11 | Extremely Low | 0.25 | 11 | Extremely Low |
| | Lianyungang | 0.22 | 12 | Extremely Low | 0.24 | 12 | Extremely Low | 0.23 | 12 | Extremely Low |
| | Suqian | 0.21 | 13 | Extremely Low | 0.23 | 13 | Extremely Low | 0.22 | 13 | Extremely Low |

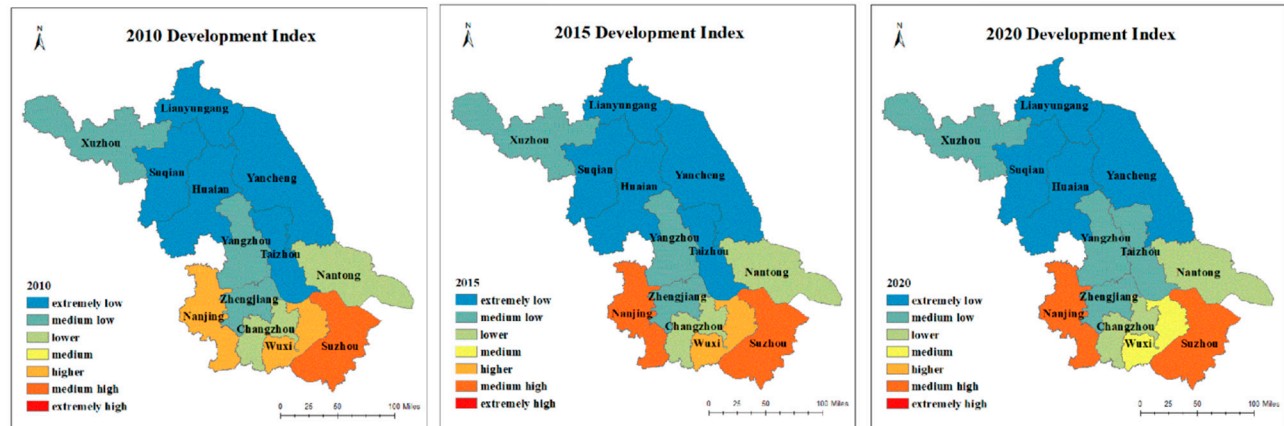

**Figure 2.** Distribution of the PES-system-development indices of each prefecture-level city in Jiangsu Province from 2010 to 2020.

In 2010, Suzhou had the highest level of development at 0.71, which indicates medium–high development. The second and third ranked cities were Nanjing and Wuxi, with values of 0.65 and 0.63, respectively, which indicate high development levels. Moreover, all three of these cities are located in Southern Jiangsu. In Central Jiangsu, except for the city of Taizhou, which has an extremely low development level, Nantong and Yangzhou have medium–low and lower development levels, respectively. In Northern Jiangsu, except for Xuzhou, which is at the medium–low development level, the remaining four cities have an extremely low level of development. In 2010, the development of the three regions was extremely uneven, with a large development gap. Generally, Southern Jiangsu has a significantly higher level of development than that of the Central Jiangsu and Northern Jiangsu.

During the 12th Five-Year Plan period, Jiangsu Province proposed the development goal of "promoting scientific development and building a better Jiangsu." After the 12th Five-Year Plan period, the city with the highest level of development according to the PES system was still Suzhou in 2015, with a score of 0.77, which was six percentage points higher than that of 2010 and remains at the middle–high development level. Nanjing had a

score of 0.70, which is five percentage points higher than its score in 2010, taking it from the higher development level to the middle–high development level. Wuxi had a score of 0.60, which is three percentage points lower than that of its score 2010 but is still within the middle–high development level. Overall, the development level of Southern Jiangsu Province is higher than that of other areas within Jiangsu. The development level in cities in Central and Northern Jiangsu is the same as that in 2010, which was not high overall. Thus, during the 12th Five-Year Plan period, while Jiangsu Province grew overall, the level of economic development within the province was still unbalanced due to the strong level of development in Southern Jiangsu but the relatively weak development in Central and Northern Jiangsu. In particular, four cities with extremely low development levels exist in Northern Jiangsu.

During the 13th Five-Year Plan period, Jiangsu Province proposed the "Strong, Rich, Beautiful, and High" development agenda, which specifically refers to "Strong Economy," "Rich People," "Beautiful Environment," and "High Social Civilization." After the development of the 13th Five-Year Plan period, the city with the highest development level in the PES system was still Suzhou, and the city with the next highest development level was Nanjing in 2020; both of these cities had medium–high development levels. Taizhou in the central region of Jiangsu Province has risen from an extremely low level of development to a medium–low level of development, which means that there are no remaining cities in Central Jiangsu with an extremely low level of development. In addition, in Northern Jiangsu, there remains four cities with extremely low levels of development. Overall, the spatial distribution of the PES-development indices in Jiangsu Province is somewhat stable. The areas with higher levels of development are mainly in Southern Jiangsu. The number and spatial distribution of cities with an extremely low level of development, which are mainly concentrated in Northern Jiangsu, remained relatively consistent over the 2010–2020 period.

### 4.3. Coordination Indices

According to the system coupling-coordination-degree calculation formula, the PES coupling-coordination degree values of 13 cities in Jiangsu Province in 2010, 2015, and 2020 were derived, and the rankings are shown in Table 8. Combined with the distribution map of the PES system coupling-coordination-degree indices of each city in Jiangsu Province from 2010 to 2020 (Figure 3), it can be seen that the coupling-coordination-degree value of Suzhou was the highest in 2010, at 0.84, which is within the well-coordinated development stage. Therefore, Nanjing and Wuxi were both in the intermediate coordinated stage of development. In Southern Jiangsu, one city was in the stage of well-coordinated development, two cities were in the stage of intermediate coordinated development, and two cities were in the stage of basic coordinated development. In Central Jiangsu, one city was in the stage of basic coordinated development (Nantong), and two cities were in the stage of barely coordinated development. In Northern Jiangsu, one city was in the stage of barely coordinated development (Xuzhou), and four cities were in the stage of fragile coordinated development. Accordingly, the coupled coordination of population, economic, and social development systems in a city is strongly and positively related to the city's overall development level. After the 12th and 13th Five-Year Plans, Suzhou had the highest coupling-coordination degree of 0.87 in 2020, which was three percentage points greater than that in 2010, and was in a well-coordinated development stage. The coupling-coordination degree value of Nanjing was 0.85. Compared with 2010, it improved by six percentage points and is in the stage of well-coordinated development. In Central Jiangsu, Nantong was still at the primary coordination stage, with a coupling-coordination value of 0.63, while Taizhou and Yangzhou were at the barely coordinated stage. Nantong benefited from the influence of the Shanghai Economic Circle, and the coordinated development of population, economy, and society was better than that of the other two cities. In Northern Jiangsu, Xuzhou was in the barely coordinated stage, and Yancheng improved from the fragile coordinated stage in 2010 to the barely coordinated stage in 2020, which

was the result of rapid development within the city in the past few years. The remaining cities in Northern Jiangsu (Huai'an, Lianyungang, and Suqian) were still in the fragile coordination stage.

**Table 8.** Ranking of PES system coupling-coordination degrees of various cities in Jiangsu Province.

| Regions | | 2010 | | | 2015 | | | 2020 | | |
|---|---|---|---|---|---|---|---|---|---|---|
| | | Value | Level | Rankings | Value | Level | Rankings | Value | Level | Rankings |
| Southern Jiangsu | Suzhou | 0.84 | Well | 1 | 0.87 | Well | 1 | 0.87 | Well | 1 |
| | Nanjing | 0.79 | Intermediate | 2 | 0.83 | Well | 2 | 0.85 | Well | 2 |
| | Wuxi | 0.79 | Intermediate | 3 | 0.77 | Intermediate | 3 | 0.77 | Intermediate | 3 |
| | Changzhou | 0.67 | Basic | 4 | 0.67 | Basic | 4 | 0.67 | Basic | 4 |
| | Zhenjiang | 0.60 | Basic | 6 | 0.62 | Basic | 6 | 0.60 | Basic | 6 |
| Central Jiangsu | Nantong | 0.63 | Basic | 5 | 0.65 | Basic | 5 | 0.63 | Basic | 5 |
| | Yangzhou | 0.59 | Barely | 7 | 0.58 | Barely | 8 | 0.57 | Barely | 7 |
| | Taizhou | 0.53 | Barely | 9 | 0.53 | Barely | 10 | 0.55 | Barely | 8 |
| | Xuzhou | 0.57 | Barely | 8 | 0.60 | Basic | 7 | 0.55 | Barely | 9 |
| Northern Jiangsu | Yancheng | 0.49 | Fragile | 10 | 0.54 | Barely | 9 | 0.51 | Barely | 10 |
| | Huai'an | 0.48 | Fragile | 11 | 0.52 | Barely | 11 | 0.49 | Fragile | 11 |
| | Lianyungang | 0.45 | Fragile | 12 | 0.48 | Fragile | 12 | 0.47 | Fragile | 12 |
| | Suqian | 0.45 | Fragile | 13 | 0.48 | Fragile | 13 | 0.46 | Fragile | 13 |

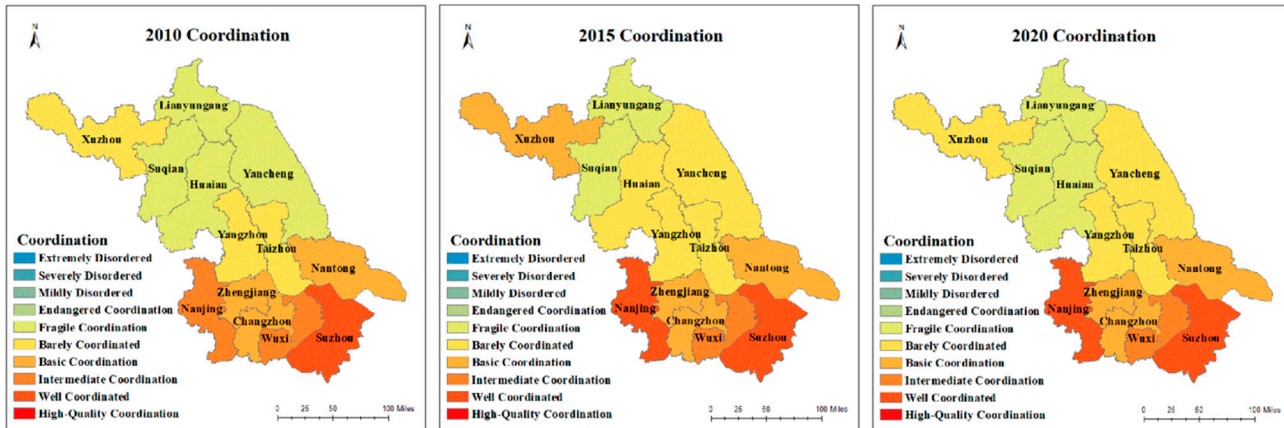

**Figure 3.** Distribution of the PES system coupling and coordination indices of each district-level city in Jiangsu Province from 2010 to 2020.

Achieving a balanced regional development in Jiangsu Province faces two problems. First, the regional development of Jiangsu Province is unbalanced, and there are clear coordinated development blocks in Southern Jiangsu around Suzhou and Nanjing, barely coordinated development clusters in Central Jiangsu, and fragile coordinated development centers in Northern Jiangsu. Second, progress in closing the development gap in Jiangsu Province has been slow. Only Nanjing progressed from the intermediate coordination stage in 2010 to the well-coordinated coordination stage in 2020, and Yancheng improved from the fragile coordination stage to the barely coordinated stage over the same period. Otherwise, there was no change in the coupled coordination status of the remaining cities in the region.

## 5. Discussion

The balanced and high-quality development of urban agglomerations is an important component and the foundation for building a new development pattern [51]. Thus, it is crucial to explore the reasons for the uneven development of urban agglomerations and discover countermeasures to establish new high-quality development patterns. The present study showed that the uneven development of city clusters in Jiangsu Province is mainly due to the following four reasons.

### 5.1. Geographic Differences

Southern Jiangsu belongs to the core of the Yangtze River Delta Economic Circle. Influenced by the Shanghai area, since the 1980s, Southern Jiangsu vigorously developed township and village enterprises, established development zones, and attracted international investment [52]. These changes led to industrial and technological transfer throughout the Shanghai area, which was a major catalyst for economic development in Southern Jiangsu. This is a significant driving factor that enabled Southern Jiangsu to be in the leading position in the province in terms of economic development. Central Jiangsu has also achieved a certain degree of development under the influence of the Nanjing Metropolitan Area and the Yangtze River Delta Economic Belt [53]. However, compared with Southern Jiangsu, there remains room for improvement in terms of economic development. Northern Jiangsu is far from Shanghai, Nanjing, and other economic centers, so geography is a disadvantage. There are no large and thriving cities to drive development [54], and development is lagging behind that of other cities. Geographic differences are a significant factor in the uneven regional development in Jiangsu Province.

### 5.2. The Matthew Effect

The Matthew effect is a theory of the accumulation of dominance that is used to describe the phenomenon of "the strong becoming stronger and the weak becoming weaker" [55]. This theory has been widely applied to research in various disciplines [56]. The "Matthew effect," when applied to regional development, refers to the fact that regions with a high level of regional development will continue to grow, which means that the level of development in Southern Jiangsu will always be higher than that in Central Jiangsu and Northern Jiangsu [57]. Since the reform and opening up, Southern Jiangsu has accumulated wealth that is incomparable to that of Central Jiangsu and Northern Jiangsu and has formed a solid economic foundation of technology, capital, and talent. This solid good foundation provides the necessary conditions for all kinds of innovation [58], which then attracts more resources, creating a "siphon" phenomenon where Southern Jiangsu absorbs more advanced technology, talented people, and high-quality capital, accelerating further development [59]. The economic foundation of Northern Jiangsu has always been inferior to that of Southern Jiangsu. Therefore, this area is unable to attract production factors; thus, its development continues to lag behind. The difference in economic foundations has led to the Matthew effect in regional development, which exacerbates the unevenness of regional development.

### 5.3. Game Thinking

Economic development is, to some extent, linked to the performance of local governments. In practice, local authorities tend to maximize their own interests [60], and economically developed regions may believe that "gaming" is preferable to "mutual benefit" [61]. This leads to a lack of mobility for production factors, and inter-regional links are weak, which hinders the formation of effective cooperation mechanisms. Thus, these inter-regional barriers make it difficult to achieve balanced development within Jiangsu Province. Game thinking is one plausible explanation for the unbalanced regional development in Jiangsu Province [62].

### 5.4. Differences in Industrial Structure

Industrial structure refers to the proportional relationship between industries in the national economy and the connections between them [63]. According to data from the Jiangsu Provincial Statistical Yearbook, the proportion of the output value of primary and secondary industries in each region of Jiangsu Province from 2010 to 2020 decreased over time while the proportion of the output value of tertiary industry increased. However, in terms of industrial structure, the proportion of the output value of primary and secondary industries in Southern Jiangsu is much smaller than that in Central and Northern Jiangsu, while the proportion of the output value of tertiary industry is much greater than other

areas [64]. The difference in industrial structure has caused different speeds of economic development between the three major regions of Jiangsu Province [65], which have resulted in unbalanced regional economic development.

## 6. Conclusions

In summary, this research paper has investigated the coordinated development of smart-city clusters in Jiangsu Province, China, using a multidimensional approach focusing on population, economy, and society. By constructing an evaluation index system tailored to these dimensions and employing a coupling-coordination degree model for quantitative analysis, several key findings have emerged regarding the coordination within the city clusters. The application of the coupling-coordination degree model has revealed important insights into the level of coordination among different aspects of urban development within the smart-city clusters. It has highlighted areas of synergy and alignment, as well as areas of potential discord or inefficiency. Specifically, the findings suggest varying degrees of coordination across the population, economy, and social dimensions, with certain factors exhibiting higher levels of coupling than others. This indicates both strengths and areas for improvement in the coordinated development of smart-city clusters in Jiangsu Province. The development and coordination indices of the three regions in Jiangsu Province were analyzed in depth. There are clear differences in the development levels of the three regions, with the overall development level of Southern Jiangsu being higher than that of Central and Northern Jiangsu. In terms of the degree of coupling and coordination, the overall development of Southern Jiangsu is better than that of Central and Northern Jiangsu.

Throughout the history of economic and social development in China and the West, due to different factors, such as location, resource endowment, productivity level, and social system, the problem of uneven regional development has existed for a long time. This problem should be addressed in a scientific manner [66]. Given the context of smart cities developing worldwide, smart-city clusters will ultimately form. To increase the efficiency of such development [67], it is important to balance the development between city clusters. To foster coordinated development among smart cities and city clusters within a region, this study proposes the following recommendations.

### 6.1. Strengthen Cross-Regional Cooperation

Strengthening regional cooperation is important to ensure balanced development between city clusters [68]. Therefore, it is necessary to clearly define and plan development strategies for each city, break down geographical barriers, and deepen cooperation among cities in the region. By facilitating the flow of production factors and effectively integrating various types of resources, each region can leverage their respective comparative advantages and maximize their benefits through an optimal division of labor. Moreover, deepening cross-regional cooperation between city clusters can solve common problems in building smart cities [69]. Smart cities should share their successful experiences and best practices, specify their goals and areas of cooperation, and collectively explore their common challenges and solutions in the construction of smart cities. By strengthening cooperation among cities, the duplication of resources and waste can be avoided, raising the overall efficiency of smart-city construction, respective development capacities, and mutual benefits [70].

### 6.2. Promote Data Sharing and Interoperability

The construction of smart cities involves the collection and processing of a large amount of data [71], including transportation data, environmental data, and energy data. Thus, data sharing and interoperability are important components of creating smart cities. Cities can establish a data-sharing platform to achieve data interoperability and sharing. The data-sharing platform can include functions such as data collection, storage, processing, and visualization [72]. Cities can share their data resources with other cities through this platform, improving the efficiency of data utilization. In addition, data sharing can promote

exchanges and cooperation between cities and foster synergistic development [73]. This approach can better utilize data resources and improve the efficiency of urban management and decision-making.

### 6.3. Deepen Industrial Synergistic Development

The synergistic development of industries among city clusters is important for the construction of smart cities, which involves the integrated development of several industries, including transportation, energy, and the environment. City clusters within the same region can strengthen cooperation in these sectors to create industrial synergy [74]. For example, in transportation, cities can jointly develop intelligent traffic-management systems to share traffic information and improve the efficiency of traffic management [75]. In energy, cities can collectively promote clean energy to reduce energy consumption and pollution emissions [76]. In environmental protection, cities can work together to carry out environmental monitoring and pollution control [77]. The competitiveness of smart-city construction can be enhanced through collaborative industrial development.

### 6.4. Foster Innovation Capacity

Enhancing urban innovation capacity is key to the construction of smart cities [78], which requires innovative thinking and technology [79]. Thus, city clusters within the same region can create and strengthen innovation mechanisms and institutions to cultivate innovation capacity. Specifically, cities should increase their investment in science and technology innovation to support the relevant enterprises and scientific research institutions. Cities should also strengthen the development of talent, establish a knowledge base for smart-city construction, and cultivate a group of smart-city construction professionals with innovative mindsets and practical skills. In addition, cities should adopt a healthy talent-management system to absorb and retain talent. Furthermore, cities should increase cooperation with international advanced science and technology enterprises and research institutions as doing so can enhance urban innovation capacity [80].

Innovation and Limitations:

This research contributes significantly to the field in several ways. Firstly, it addresses the pressing need for coordinated development in smart-city clusters, which is crucial for maximizing the benefits of urbanization and technological advancement. By focusing on Jiangsu Province, a key economic and technological hub in China, the study provides valuable insights that can be applied to other regions facing similar challenges. Secondly, the construction of a comprehensive evaluation index system allows for a more nuanced understanding of the factors influencing the development of smart-city clusters. By considering population dynamics, economic indicators, and social factors, the study provides a holistic framework for assessing the progress and identifying areas for improvement in urban planning and governance. Thirdly, the use of a coupling-coordination degree model for quantitative analysis represents a methodological innovation in the field of urban studies. This approach enables researchers and policymakers to not only measure the level of coordination within city clusters but also to identify the factors driving or hindering this coordination. By quantifying the relationships between different variables, the model provides valuable insights that can inform decision making and policy formulation. Overall, this research makes a significant contribution to the understanding and promotion of coordinated development in smart-city clusters, with implications for urban planning, governance, and policy formulation in China and beyond. Its interdisciplinary approach, combining insights from urban studies, economics, and sociology, sets a valuable precedent for future research in this rapidly evolving field.

While the research on the coordinated development of smart-city clusters in Jiangsu Province contributes valuable insights, it is important to acknowledge its limitations. Data Availability and Quality: The reliability and availability of data can significantly affect the accuracy and robustness of the research findings. While efforts are made to gather relevant data for analysis, there may be limitations in terms of data completeness or timeliness.

Limited Scope of Evaluation Indicators: Despite efforts to construct a comprehensive evaluation index system covering population, economy, and society, there may be other relevant factors not included in the analysis. For instance, environmental sustainability, infrastructure quality, or innovation capacity could also play significant roles in shaping the development of smart-city clusters but may not be adequately captured by the selected indicators. Acknowledging these limitations is crucial for interpreting the findings of the research accurately and for guiding future studies to address these gaps, thereby advancing our understanding of the coordinated development of smart-city clusters in China and beyond.

Future planned work that can be carried out: Conducting longitudinal studies to track the evolution of smart-city clusters over time would offer deeper insights into their development dynamics. This could involve examining trends in population growth, economic indicators, and social factors, and identifying patterns of change and continuity.

**Author Contributions:** B.L. and G.S. contributed to the study conception and design. Material preparation, data collection and analysis were performed by T.Y. and K.Z. The first draft of the manuscript was written by B.L. Z.S. revised and improved manuscript. All authors have read and agreed to the published version of the manuscript.

**Funding:** This work was supported by the Fundamental Research Funds for the Central Universities: Climate Migration Types and Risk Management in Coastal Areas. (Grant number B230205032); & Postgraduate Research & Practice Innovation Program of Jiangsu Province: Climate Migration Types and Risk Management in Coastal Areas. (Grant number: 422003151); and The Key Research Project of the National Foundation of Social Science of China: Community Governance and Post-relocation Support in Cross District Resettlement [Grant number 21&ZD183].

**Institutional Review Board Statement:** The local Ethics Committee of Hohai University approved the consent form.

**Data Availability Statement:** The original contributions presented in the study are included in the article, further inquiries can be directed to the corresponding author.

**Conflicts of Interest:** The authors declare no conflicts of interest.

## Notes

[1] There are various studies on the connotation of social development. In this paper, social development refers to the ecological environment, infrastructure, social security system, and scientific and educational development of the whole society, excluding population development and economic development. Essentially, social development is the social attributes of the environment and resources. The study of a population–economic–social-development system in this paper belongs to the research subfield of population economics.

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
