# Peer review of "Exploring the Coordinated Development of Smart-City Clusters in China: A Case Study of Jiangsu Province"

_land, doi:10.3390/land13030308_

Round 1
Reviewer 1 Report
Comments and Suggestions for Authors
- Research problem statement has some relation to the problem, but is either incomplete, poorly expressed or tangential; Aims and objectives are somewhat relevant, but are either not achievable or do not completely follow from the research problem; Background to the research problem indicates some awareness of the issues, but misses key issues or points and the context is underspecified.
-A reasonable overview of the subject is provided, based on published work; The discussion identifies relevant principles and/or approaches relevant to the research problem and displays a basic understanding of them; The scope of references is standard and acceptable.
-The selection of research design and methods is appropriate and is clearly presented. however, the relevance of the methodology for the research problem and objectives is not adequately justified; There is little to no reference to the limitations of data sources.
-The research design is fairly consistently applied in the production of data; Research findings/evidence are classified in clear categories which are basically relevant to the research problem - occasional inconsistencies or lack of clarity does not detract from the overall presentation and argument; The research findings/evidence are clearly communicated and reasonably logically presented.
-The discussion is soundly reasoned and relates the findings to the broader context of the work; A reasonable attempt is made to interpret results in a critical light, noting shortcomings and significant achievements and taking account of obvious factors; The discussion identifies one or more clear arguments, supports them with evidence and links them to the overall context of the work; Key points in the evidence/findings are identified and their implications for the research problem, objectives and the literature are discussed; A clear set of conclusions address most of the objectives.
- The authors must include implications to the field and novelty of the research.
Comments on the Quality of English Language
- Research problem statement has some relation to the problem, but is either incomplete, poorly expressed or tangential; Aims and objectives are somewhat relevant, but are either not achievable or do not completely follow from the research problem; Background to the research problem indicates some awareness of the issues, but misses key issues or points and the context is underspecified.
-A reasonable overview of the subject is provided, based on published work; The discussion identifies relevant principles and/or approaches relevant to the research problem and displays a basic understanding of them; The scope of references is standard and acceptable.
-The selection of research design and methods is appropriate and is clearly presented. however, the relevance of the methodology for the research problem and objectives is not adequately justified; There is little to no reference to the limitations of data sources.
-The research design is fairly consistently applied in the production of data; Research findings/evidence are classified in clear categories which are basically relevant to the research problem - occasional inconsistencies or lack of clarity does not detract from the overall presentation and argument; The research findings/evidence are clearly communicated and reasonably logically presented.
-The discussion is soundly reasoned and relates the findings to the broader context of the work; A reasonable attempt is made to interpret results in a critical light, noting shortcomings and significant achievements and taking account of obvious factors; The discussion identifies one or more clear arguments, supports them with evidence and links them to the overall context of the work; Key points in the evidence/findings are identified and their implications for the research problem, objectives and the literature are discussed; A clear set of conclusions address most of the objectives.
- The authors must include implications to the field and novelty of the research.
Author Response
Thank you for taking the time to review our manuscript and providing suggestions for revisions. We have completed the manuscript revisions as required.
(1)Research problem statement has some relation to the problem, but is either incomplete, poorly expressed or tangential; Aims and objectives are somewhat relevant, but are either not achievable or do not completely follow from the research problem; Background to the research problem indicates some awareness of the issues, but misses key issues or points and the context is under specified.
Reply:We have rewritten the research background, as detailed in the introduction section of Chapter 1.
(2)The selection of research design and methods is appropriate and is clearly presented. however, the relevance of the methodology for the research problem and objectives is not adequately justified; There is little to no reference to the limitations of data sources.
Reply:We discussed in Chapter 3.2 why the coupled coordination development model was adopted. In the second paragraph of Chapter 2, it discusses the establishment of an evaluation index system from three dimensions: population, economy, and society.The issue of data limitations was further explained at the end of the paper.
(3)The authors must include implications to the field and novelty of the research.
Reply:In the Conclusions section of Chapter 6, the implications and innovation of this research were supplemented.

Reviewer 2 Report
Comments and Suggestions for Authors
Dear Authors,
Unfortunately, I wasn't able to find many references, may be the titles were translated into English. May be you can provide the doi? It is very hard to provide the opinion without checking all the references. More international works should be analysed and referenced.
Despite of this I have found some weak points:
Line 34-35 more references should be provided for smart city definition perspectives. For example https://commission.europa.eu/eu-regional-and-urban-development/topics/cities-and-urban-development/city-initiatives/smart-cities_en
The limitations of the research should be clearly presented at the end of the manuscript.
Why have you chosen just a set of concrete factors and avoid many of them? Here you have referenced the sources, but didn't explain reasons clearly. Please provide more extensive explanation (line 135-136)
You should better describe the formula's sources or better explain your contribution. You provided the source [46], but the formulas in this paper are different. For formulas [3] and [4] it will be better to provide the source [47].
Line [63] please provide the source of the definitions.
Line 200 - it will be better to provide the abbreviation of the entropy weight (e).
Please provide better explanation to equations 11 and 12.
The bibliography positions should be better adapted to MDPI style.
To sum up the main body of the paper, the main idea should be better reasoned, more references should be provided, the equations should be better described, the local references perspective should be enlarged.
The recommendations, in my opinion, should have more scientific manner, less the political one.
You should also describe the authors contribution, as well as which future works are (can be) planned.
Have you established concrete hypotheses at the beginning of research? They should be presented at the introduction and analysed at the end of the research.
It will be significant to describe in conclusion will this "approach" be able to be applied to other countries or is it only of local importance? Is there any perspective of copying it?
Regards.
Comments on the Quality of English LanguageMinor editing of English language required.
Author Response
Thank you for taking the time to review our manuscript and providing suggestions for revisions. We have completed the manuscript revisions as required.
(1).Line 34-35 more references should be provided for smart city definition perspectives. For example https://commission.europa.eu/eu-regional-and-urban-development/topics/cities-and-urban-development/city-initiatives/smart-cities_en
Reply:We have added more references to smart cities
(2).The limitations of the research should be clearly presented at the end of the manuscript.
Reply:We have added limitations to the research at the end of the manuscript.
(3).Why have you chosen just a set of concrete factors and avoid many of them? Here you have referenced the sources, but didn't explain reasons clearly. Please provide more extensive explanation (line 135-136)
Reply:We have constructed an evaluation index system from three dimensions: population, economy, and society, which is a first level evaluation index. For each specific primary indicator, it is further subdivided into secondary indicators. We have selected secondary indicators for an overall overview and introduced references. The specific three-level indicators are determined through the expert method.
(4).You should better describe the formula's sources or better explain your contribution. You provided the source [46], but the formulas in this paper are different. For formulas [3] and [4] it will be better to provide the source [47].
Reply:This section is an introduction to the entropy weight method, which is an internationally standardized formula. The reference 46 cited here refers to the scientific introduction of the entropy weight method in this paper. Nevertheless, we have replaced reference 46, which contains relevant formula descriptions.
(5).Line [63] please provide the source of the definitions.
Reply:The source of the definition can be found in reference 14
(6).Line 200 - it will be better to provide the abbreviation of the entropy weight (e).
Reply:Entropy Weighting Method Abbreviation (EWM) provided
(7).Please provide better explanation to equations 11 and 12.
Reply:Formula 11 is the development index, Formula 12 is the coordination index, which is a universal formula for coupling coordination models. See references 48 to 50.
(8).The bibliography positions should be better adapted to MDPI style.
Reply:During the manuscript proofreading process, we will work with the editorial department to determine the reference location for the bibliography.
(9)To sum up the main body of the paper, the main idea should be better reasoned, more references should be provided, the equations should be better described, the local references perspective should be enlarged.
Reply:We have supplemented the summary section and references of the manuscript.
(10).The recommendations, in my opinion, should have more scientific manner, less the political one.
Reply:On the basis of the evaluation of coordinated development of urban agglomerations in Jiangsu Province, we have collected suggestions from relevant experts and government officials and proposed suggestions for coordinated development of urban agglomerations. We have sublimated the suggestions, taking into account their scientificity and reference value.
(11).You should also describe the authors contribution, as well as which future works are (can be) planned.
Reply:The author's contributions have specific areas for discussion. We have added the next step of the work plan in the last paragraph of the article.
(12).Have you established concrete hypotheses at the beginning of research? They should be presented at the introduction and analysed at the end of the research.
Reply:We did not use regression methods, so the manuscript does not include research hypotheses.
(13).It will be significant to describe in conclusion will this "approach" be able to be applied to other countries or is it only of local importance? Is there any perspective of copying it?
Reply:We have provided a detailed explanation of the applicability and innovation of our research methods in the conclusion section of the manuscript.

Round 2
Reviewer 2 Report
Comments and Suggestions for Authors
Dear Authors,
Still remain some small misunderstandings, but generally, the corrections made are satisfactory enough for me to accept the paper in present form.
The reference 7 doesn't work, please check and correct it.
Comments on the Quality of English LanguageThe English is ok, small revisions could be done.